# Exploring the Impact of Exercise-Derived Extracellular Vesicles in Cancer Biology

**DOI:** 10.3390/biology13090701

**Published:** 2024-09-06

**Authors:** Monica Silvestri, Elisa Grazioli, Guglielmo Duranti, Paolo Sgrò, Ivan Dimauro

**Affiliations:** 1Unit of Biology and Genetics of Movement, Department of Movement, Human and Health Sciences, University of Rome Foro Italico, 00135 Rome, Italy; m.silvestri5@studenti.uniroma4.it; 2Unit of Physical Exercise and Sport Sciences, Department of Movement, Human and Health Sciences, University of Rome Foro Italico, 00135 Rome, Italy; elisa.grazioli@uniroma4.it; 3Unit of Biochemistry and Molecular Biology, Department of Movement, Human and Health Sciences, University of Rome Foro Italico, 00135 Rome, Italy; guglielmo.duranti@uniroma4.it; 4Unit of Endocrinology, Department of Movement, Human and Health Sciences, University of Rome Foro Italico, 00135 Rome, Italy; paolo.sgro@uniroma4.it

**Keywords:** physical activity, biological mechanisms, cancer, extracellular vesicles, tumor growth

## Abstract

**Simple Summary:**

Cancer continues to be a major medical challenge, highlighting the need for new treatment strategies. One promising area of research is the role of exercise-derived extracellular vesicles, which are tiny particles released from cells during physical activity. These extracellular vesicles play a key role in cell communication and can influence various cellular functions. Recent studies have shown that extracellular vesicles released during exercise contain bioactive molecules that may help combat cancer. These molecules have been found to inhibit tumor growth, prevent the spread of cancer, and improve responses to treatment. They work by modulating important signaling pathways and altering the tumor environment, which could enhance the effectiveness of cancer therapies and minimize side effects. This review aims to summarize the current understanding of how exercise-derived extracellular vesicles and their contents impact cancer biology. It will cover how these extracellular vesicles affect cancer cell behaviors like growth, proliferation, and invasion, and discuss the potential benefits and limitations of using exercise-derived extracellular vesicles as new cancer treatments.

**Abstract:**

Cancer remains a major challenge in medicine, prompting exploration of innovative therapies. Recent studies suggest that exercise-derived extracellular vesicles (EVs) may offer potential anti-cancer benefits. These small, membrane-bound particles, including exosomes, carry bioactive molecules such as proteins and RNA that mediate intercellular communication. Exercise has been shown to increase EV secretion, influencing physiological processes like tissue repair, inflammation, and metabolism. Notably, preclinical studies have demonstrated that exercise-derived EVs can inhibit tumor growth, reduce metastasis, and enhance treatment response. For instance, in a study using animal models, exercise-derived EVs were shown to suppress tumor proliferation in breast and colon cancers. Another study reported that these EVs reduced metastatic potential by decreasing the migration and invasion of cancer cells. Additionally, exercise-induced EVs have been found to enhance the effectiveness of chemotherapy by sensitizing tumor cells to treatment. This review highlights the emerging role of exercise-derived circulating biomolecules, particularly EVs, in cancer biology. It discusses the mechanisms through which EVs impact cancer progression, the challenges in translating preclinical findings to clinical practice, and future research directions. Although research in this area is still limited, current findings suggest that EVs could play a crucial role in spreading molecules that promote better health in cancer patients. Understanding these EV profiles could lead to future therapies, such as exercise mimetics or targeted drugs, to treat cancer.

## 1. Introduction

Cancer continues to be one of the greatest challenges in modern medicine, requiring innovative therapeutic approaches that can enhance current treatments. Recently, there has been increasing interest in the potential of exercise-derived extracellular vesicles (EVs) as promising agents for cancer therapy. Extracellular vesicles, including exosomes and microvesicles, are small membrane-bound particles secreted by various cell types, capable of carrying bioactive molecules such as proteins, lipids, and nucleic acids. Emerging evidence suggests that these EVs play crucial roles in intercellular communication and can modulate cellular functions in both physiological and pathological conditions [1,2].

Exercise has long been recognized for its numerous health benefits, including its potential to influence cancer biology [3,4]. Recent research has unveiled exercise as a potent modulator of EV secretion, with exercise-induced EVs implicated in various physiological processes such as tissue repair, inflammation modulation, and metabolic regulation [5]. Moreover, exercise-derived EVs have shown promising anti-cancer properties, including the inhibition of tumor growth, suppression of metastasis, and enhancement of treatment response [6,7].

This review aims to provide a comprehensive overview of the current understanding of the biological effects of exercise-derived circulating biomolecules, with a particular focus on the role of EV cargo, in cancer biology. We will explore the current knowledge on the biological mechanisms induced by both exercise-conditioned serum and exercise-induced EVs, as well as their impact on cancer cell features, such as growth/proliferation, cell death, migration, and invasion. Additionally, we will discuss the potential physiological relevance, the current limitations, and the future direction of the research on exercise-derived EVs as novel therapeutic agents in cancer management. By elucidating the intricate interplay between physical exercise, extracellular vesicles, and cancer, this review seeks to shed light on possible new therapeutic targets for cancer therapy and improve patient outcomes.

## 2. Extracellular Vesicles and Physical Activity

In recent years, the study of extracellular vesicles (EVs) has garnered significant attention due to their pivotal role in intercellular communication and their potential implications in various physiological and pathological processes. Among the diverse array of factors influencing EV secretion and function, physical activity (PA) has emerged as a particularly intriguing modulator [8,9,10,11]. 

Extracellular vesicles in the blood consist of a diverse group of membranous structures released by platelets, red blood cells (which together make up over 50%), other circulating cells, and the tissues surrounding cells into the extracellular environment [12]. Classified broadly into exosomes, microvesicles, and apoptotic bodies based on their biogenesis and size, EVs serve as vehicles for intercellular communication by transporting various biomolecules, including proteins, nucleic acids, lipids, and metabolites. This ability to shuttle bioactive cargo between cells underscores their significance in orchestrating diverse physiological processes, ranging from immune regulation to tissue repair and homeostasis [13]. 

Physical activity, encompassing exercise and movement, exerts multifaceted effects on cellular and systemic physiology. Emerging evidence suggests that PA modulates the secretion, concentration, as well as composition and function of EVs, thereby influencing intercellular communication and systemic responses [14]. Both acute bouts and chronic PA have been implicated in altering EV release patterns and cargo content across different cell types [5,8,15,16,17,18,19]. Table 1 summarizes the effects of physical exercise on EV dynamics/features. 

To date, the exact molecular mechanisms responsible for the induction of EV release during exercise are still not well understood. Over the course of the last few years, several putative contributors have been suggested, including lymphocyte mobilization [20], biomechanical forces like shear, tension, and compression [21], increased intracellular calcium levels [22,23], and conditions such as an acidic environment [24] and reactive oxygen species (ROS) production in muscle cells [25].

Studies show that high-intensity exercise generally leads to a temporary increase in circulating EVs [15,26,27], while moderate-intensity exercise has mixed effects, with EV levels either rising [16,28,29], remaining unchanged [8,19,30,31], or decreasing [9,32]. Exercise intensity seems to influence EV release, with factors like exercise type, timing of sample collection, and sex also affecting EV levels and characteristics. Similarly, the analysis of EV size shows for most of the published research no change in size following aerobic exercise [33,34], while others find a decrease in EV mean size in response to resistance exercise in sedentary youth with obesity [35] and in active healthy men [36], but not in women [36]. However, differences in EV studies may also stem from variations in methods used for EV isolation and quantification. Techniques like ultracentrifugation, size-exclusion chromatography, and precipitation-based methods can vary in efficiency and may co-isolate non-EV components, affecting results [27,37]. More accurate EV counts can be achieved through methods that quantify EVs directly in biofluids, such as nano-FCM or Exoview, which minimize biases [16,38].

It is known that various tissues respond to PA and release their EVs into the bloodstream following exercise. While skeletal muscle is a major tissue involved in exercise and produces EVs rich in muscle-specific proteins, most of these EVs remain within the muscle tissue [39]. Only a small fraction (1–5%) of muscle-derived EVs enter the bloodstream [39,40,41]. Other cell types, including lymphocytes, monocytes, platelets, endothelial cells, and antigen-presenting cells, are suggested to be the main contributors to increased circulating EVs during and after exercise [27]. However, more research is needed to determine how these findings apply to different exercise types, intensities, and durations.

Analysis of the protein cargo of exercise-derived EVs revealed a variety of proteins linked to key signaling pathways, such as angiogenesis, immune signaling, and glycolysis [31,42]. Additionally, several studies provided evidence of altered ncRNA cargo in EVs following exercise [31,39,43,44,45]. Functional analysis of exercise-related EVs indicated their role in cardiovascular prevention [9], protection in ischemia/reperfusion injury [16,19], hypoxia/reoxygenation assays [19], tissue remodeling [46], endothelial function [47], as well as muscle remodeling and growth [26], potentially driven by EV cargo responding to exercise stimuli. 

To date, all these findings suggest that EVs are actively released into circulation during PA and may act as mediators of various key signaling pathways involved in exercise-induced adaptation processes.

## 3. Biological Impact of Physical Activity on Cancer

Cancer is a complex and multifaceted disease characterized by uncontrolled cell growth and proliferation. Despite advances in treatment modalities, cancer remains a significant global health burden. In recent years, the role of lifestyle factors, including PA, in cancer prevention and management has garnered considerable attention, and guidelines for cancer survivors have been presented [48,49]. Mounting evidence suggests that regular PA is associated with a reduced risk of developing certain types of cancer. Epidemiological studies have consistently demonstrated an inverse relationship between PA levels and the incidence of several common cancers, including breast, colon, prostate, and lung cancer [50,51]. Moreover, emerging evidence suggests that exercise may also improve cancer outcomes and overall survival among cancer survivors [48,52,53].

The molecular mechanisms underlying the beneficial effects of exercise in cancer prevention and management are multifaceted and involve both systemic and cellular pathways [53,54,55,56]. With this understanding, exercise training for cancer patients could shift from a ‘one-size-fits-all’ approach to more personalized strategies, grounded in detailed physiological insights into how varying amounts, intensities, and types of exercise can influence cancer outcomes. To date, it is known that regular PA is associated with modulation of various physiological processes, including inflammation, immune function, hormone metabolism, and oxidative stress, all of which play critical roles in cancer development and progression [57,58,59]. Additionally, PA may directly influence intrinsic tumor factors such as growth rate, angiogenesis, apoptosis, metabolism, and metastasis [60,61]. It also helps alleviate adverse effects associated with cancer and its treatments, while enhancing the effectiveness of cancer therapies.

Among the many studies examining the impact of PA on cancer outcomes, the most frequently investigated effect is the reduction in tumor growth rate [61,62,63]. Notably, research has shown that exercise training can lead to up to a 67% reduction in the growth rate of established tumors [62].

To better understand at the biological level the growth-inhibitory effect of PA, several studies have utilized exercise-conditioned serum to incubate cancer cells from the breast, prostate, and lung [64,65,66,67,68,69,70,71,72,73]. Table 2 lists key findings from treatment of cancer cells with exercise-conditioned serum. 

To date, it is still not clear whether acute exercise is better than chronic exercise in terms of effectiveness on tumor cells, probably due to the differences in patient population, type of exercise, and cancer cell types [64,68,71,72]. The current results suggest an impact of exercise-conditioned serum on both the phosphorylation state of protein involved in signaling pathways related to proliferation (i.e., STAT3, Akt, mTOR, p70s6k, and Erk 1/2), and supporting the Hippo tumor suppressor pathway by inhibiting Yes-Associated Protein (YAP)/PDZ-binding Motif (TAZ) in different cancer cells [67,69,73] (Figure 1), known to be dysregulated in many cancers.

In 2013, Rundqvist and colleagues [66] were the first group to demonstrate that exercise-conditioned serum from healthy subjects reduces prostate cancer cell viability (LNCaP cells) by ~30% when compared with serum collected pre-exercise. Serum analysis identified two possible candidates for the effect: increased insulin like growth factor binding protein-1 (IGFBP-1) and reduced levels of epidermal growth factor (EGF) [66]. It is known that low levels of IGFBP-1 [74,75,76], as well as high levels of EGF and its receptor, are found in prostate cancer and are associated with poor prognosis [77,78]. Therefore, their exercise-induced modulation suggests a direct role of PA on the reduced rate of proliferation observed in these cancer cells (Figure 1).

Similar findings were observed with exercise-conditioned serum from comparable populations, including young healthy men [67] and older prostate cancer patients [71,72], as well as with human breast cancer cell lines (e.g., MDA-MB-231, MCF-7) treated with serum from both young healthy women [67] and women with breast cancer [64,73]. Notably, mice injected with cancer cells treated with post-exercise human serum developed fewer tumors compared to those injected with cells treated with pre-exercise serum [73].

At a mechanistic level, not all of the signaling pathways modified by exercise-conditioned serum involved in the process of inhibition of tumor proliferation are yet known; however, the results suggest a possible modulation of specific myokines (i.e., IL-6, IL-15, and oncostatin M [OSM]) and the Hippo tumor suppressor pathway in both cell lines, with a partial increase in YAP phosphorylation, and, only in breast cancer cells, the possible involvement of the Wnt/ß-catenin pathway, credited with a significant decrease in GSK3ß phosphorylation (Figure 1). Furthermore, inhibition of some of these molecules/pathways has been shown to attenuate the effect of exercise on tumor latency [79,80].

Again in human colon cancer, Devin and colleagues [68] identified a transient increase in the concentration of serum cytokines (i.e., TNFalpha, IL-6, and IL-8) immediately after high-intensity interval exercise in colorectal cancer survivors, which may be an important mechanism contributing to the observed growth suppression effect in colon cancer cells. A few years later, Orange and colleagues found similar results with exercise-conditioned serum from healthy, sedentary males [81]. Although there are differences in terms of exercise protocol (acute aerobic vs. acute high-intensity interval exercise) and the characteristics of recruited subjects (Healthy vs. Disease condition), immediately after the single bout of exercise they found a significant increase in IL-6. To date, the mechanism through which aforementioned factors could influence cell growth in vivo is not yet clear. For some of these molecules, a potential mechanism has been proposed to explain the anti-carcinogenic effects of exercise-conditioned serum, although these hypotheses have not yet been explored in the context of exercise and cancer. For instance, it is possible that interleukin-6 (IL-6) released from skeletal muscle during exercise activates AMP-activated protein kinase (AMPK) in distant tissues, including aberrant or dysplastic cells, beyond just adipocytes [82]. AMPK is known to inhibit mTOR and its downstream targets, such as p70S6K, through either a tuberous sclerosis complex 2 (TSC2)-dependent or -independent pathway [83,84] (Figure 1).

Additionally, Kurgan and colleagues found that acute exercise led to significant increases in IL-6, IL-1β, IL-1α, and TNF-α. Post-exercise serum was shown to inhibit the activation of Akt, its downstream effectors, including mTOR and p70S6K, as well as inhibit Erk1/2, contributing to reduced cell proliferation and survival in lung cancer cells (A549) [69] (Figure 1).

Overall, these findings underscore the significant biological impact of PA on cancer prevention and management. Regular PA has demonstrated the potential to reduce cancer risk and improve outcomes across several cancer types, including breast, prostate, lung, and colon cancers. These beneficial effects are mediated through complex molecular mechanisms, involving systemic changes in inflammation, immune function, and hormone metabolism, as well as direct modulation of tumor growth, angiogenesis, apoptosis, and metastasis.

Moreover, studies utilizing exercise-conditioned serum have revealed that both acute and chronic exercise can influence cancer cell signaling pathways, such as STAT3, Akt, mTOR, and the Hippo tumor suppressor pathway, resulting in reduced cancer cell proliferation and tumor growth. The modulation of specific myokines and cytokines, including IL-6 and TNF-α, further highlights the intricate role of exercise-induced systemic changes in the inhibition of cancer progression.

Despite these advancements, several questions remain unanswered. The exact mechanisms through which exercise exerts its anti-carcinogenic effects are not fully elucidated, and the relative effectiveness of different exercise regimens on cancer outcomes warrants further investigation. Additionally, personalized exercise interventions tailored to individual cancer types, stages, and patient profiles could enhance the therapeutic potential of PA. Continued research in this area will be vital in refining exercise guidelines and optimizing cancer care.

## 4. Exercise-Derived Extracellular Vesicles in Cancer

As mentioned in the previous chapter, numerous studies have suggested that some of the biomolecules that enter the bloodstream after exercise may reduce the proliferation rate and survival of tumor cells.

Most of the circulating exercise-derived EVs come from immune system cells, endothelial cells, and platelets, and they remain in the bloodstream after exercise [85]. Research into the phenotype of exercise-derived EVs has revealed that only a small percentage originate from skeletal muscle, with the majority of the cargo in these EVs being derived from various other tissue cell types, including hepatocytes, adipose cells, immune cells, and endothelial cells [42,86].

Skeletal muscle, which constitutes around 40% of body mass, functions as an endocrine organ by secreting various biomolecules (such as proteins, non-coding RNAs, lipids, and metabolites) released during muscle contraction and crucial for mediating some of the systemic effects of exercise [87,88]. Many of these biomolecules, often referred to as exerkines, are believed to circulate through the body enclosed in extracellular vesicles (EVs) [89]. Exercise has been shown to increase the release of over 300 molecules from skeletal muscle-derived EVs, including myokines, miRNAs, and glycolytic enzymes, which may act as tumor suppressors and influence several characteristics of cancer cells [5,65,90]. The skeletal muscle secretome, including myokines and miRNAs, has demonstrated potential in suppressing tumor growth [48,65,91]. However, the current shortage of preclinical studies directly examining the effects of skeletal muscle-derived EVs on cancer cells limits our understanding of the precise role of exercise-induced skeletal muscle-derived EVs.

To date, research on exercise-derived EVs in cancer is still in its early stages, and further studies are needed to fully understand their role. Currently, only one preclinical study has shown a direct tumor-suppressive effect of exercise-derived EVs (7) (Figure 2).

The authors found that regular injections of exercise-induced extracellular vesicles (EVs) in tumor-bearing rats led to a reduction in primary tumor growth by approximately 35% and potentially delayed the onset of lung metastases [7]. Analysis of the EV cargo revealed an upregulation of genes encoding proteins involved in metabolic processes, such as Notum (palmitoleoyl-protein carboxylesterase), Pctp (phosphatidylcholine transfer protein), and Cyp4b1 (cytochrome P450, family 4, subfamily b, polypeptide 1). Additionally, molecular chaperones such as DnaJ Heat Shock Protein Family (Hsp40) Member B5 (Dnajb5) and Heat Shock Protein Family A (Hsp70) Member 5 (Hspa5), which are crucial for protein maturation and cell survival under stress, were identified. The cargo also included molecules linked to inflammation, such as Leukotriene B4 receptor 2 (Ltb4r2) and Arachidonate 5-lipoxygenase (Alox5), T-cell development (Zinc Finger And BTB Domain Containing 1, Zbtb1), cellular response to hormones (Oxytocin receptor, OXTR), and nucleic acid metabolism (Decapping exoribonuclease, DXO). Despite the small sample size, this study provides insights into the molecular composition of exercise-induced EVs and their potential direct role in inhibiting tumor growth and metastasis in vivo.

## 5. Physiological Relevance and Limitations

Before fully embracing the potential of translating these findings into clinical practice, it is crucial to evaluate how accurately in vitro models represent real physiological conditions. A key question is whether a tumor growing or developing in vivo will be exposed to the active EV molecules that demonstrate a suppressive effect on growth, as observed in vitro.

In vivo, this will depend on a mix of factors, including the degree of vascularization and perfusion of the tumor, as well as whether the active proteins or metabolites can readily cross the endothelial barrier and infiltrate the interstitial fluid surrounding the tumor.

In light of published studies, it is clear that this question cannot be answered. Indeed, we must consider some possible limitations related to how representative the in vitro system is, especially if based on 2D monocultures of tumor cells grown on flat plastic surfaces, of the pathophysiological scenario. For example, in vitro, it is not possible to analyze some in vivo aspects that could compromise the anti-tumor effect of these exercise-induced circulating molecules such as tumor vascularization and perfusion, as well as tissue architecture and tumor microenvironment, including the extracellular matrix, cell-to-cell contacts, fibroblasts, and other stromal cells associated with the tumor that are absent. Furthermore, most cell lines employed to date may fail to predict tumor response in vivo due to their limited ability to provide inter-tumoral and intra-tumoral heterogeneity. Last but not least, it remains unclear which type of training mode (i.e., resistance training vs. aerobic training), along with associated volume and intensity, is more effective in altering EV cargo and circulating factors, which then have suppressive effects on cancer cells.

## 6. Conclusions and Future Directions

EV cargo can be considered a possible mechanism for the beneficial effects of exercise on cancer. The limited number of studies available in a relatively new area of exercise oncology precludes definitive conclusions from being drawn. To date, the results suggest a possible central role of EVs in spreading exercise-induced active molecules, such as proteins and ncRNAs, modulating a healthier global condition in cancer patients. By understanding and exploiting the specific profiles of EVs released during exercise, it might be possible to create future innovative therapeutic strategies based on exercise mimetics that deliver these benefits in vivo and/or identify specific molecular targets of drugs for new oncological therapies.

Our knowledge in this field is still in its early stages, and many crucial questions remain unanswered. As a result, due to the speculative nature of exercise-derived EVs at this point, further in vitro and in vivo research is necessary.

In vitro studies could provide insight into the intricate molecular and exercise-induced regulatory mechanisms that control the release of EVs and their role in regulating cancer cell behavior, including proliferation, apoptosis, and metastasis. These studies should also include the application of a 3D assay, co-culture systems, and/or the use of scaffold mimicking the extracellular matrix, as well as a focus on evaluating whether these EVs can enhance the effects of chemotherapy, radiation, or immunotherapy. These combination approaches could lead to more effective and less toxic treatment regimens.

In vivo studies could explore the following: (1) how these EVs alter the immune landscape, potentially enhancing anti-tumor immunity or reducing immunosuppressive signals; (2) the optimal exercise regimen for producing beneficial EVs; (3) if the therapeutic effects of EVs are durable and if any long-term adverse effects emerge; and (4) the potential toxicity of administering these EVs. Finally, studies including patient-derived xenograft (PDX) models are recommended to understand the effects of exercise-derived EVs in a more clinically relevant context, potentially accelerating the translation of research into human trials.

## Figures and Tables

**Figure 1 biology-13-00701-f001:**
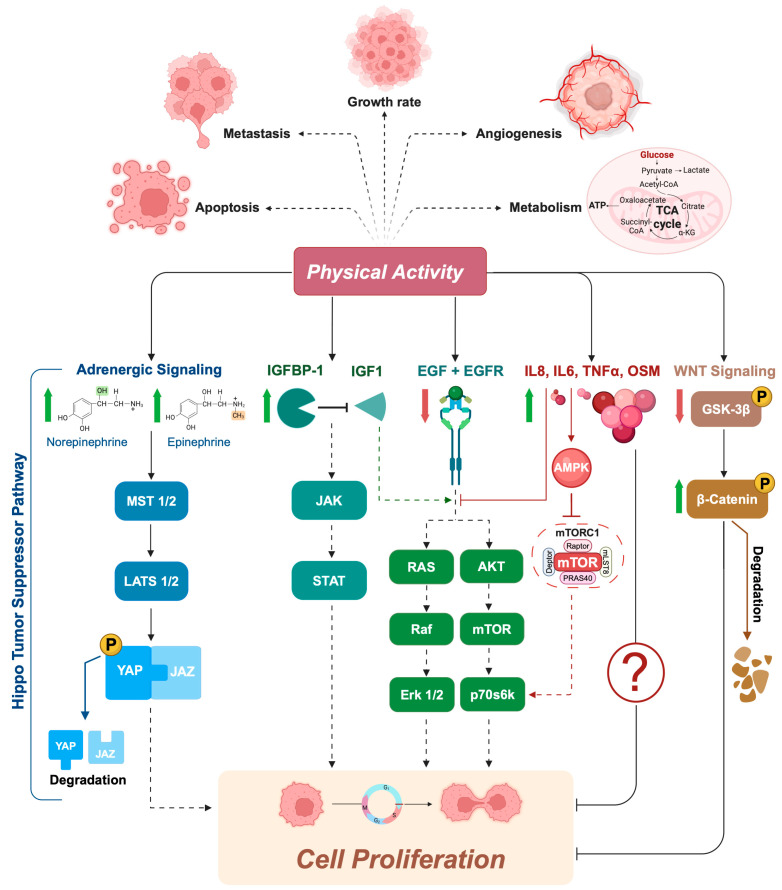
Exercise-conditioned serum inhibits signaling pathways that are crucial for cell proliferation. In cancer cells, increased proliferation is often driven by mutations in highly conserved signaling networks that regulate cell growth and division. When cancer cells are exposed to exercise-conditioned serum, several of these signaling pathways are altered, leading to a reduction in cell proliferation. For example, post-exercise serum has been shown to be enriched in catecholamines (e.g., norepinephrine and epinephrine), which can support the Hippo tumor suppressor pathway. When the Hippo pathway is “ON”, MST1/2 and MAP4K are activated, which subsequently phosphorylate and activate LATS1/2 kinases. Activated LATS1/2 phosphorylates transcriptional coactivator YAP/TAZ, preventing entry into the nucleus by promoting their degradation in the cytoplasm. Exercise has also been shown to increase the concentration of IGFBP1, a protein known to regulate cell proliferation, survival, differentiation, migration, and invasion. The binding of IGFBP1 with IGF1 modulates the activity of IGF1 signaling axes, such as the JAK, RAS, and AKT pathways, by regulating their availability to the IGF-IR. Similar effects were found with lower levels of EGF in exercise-conditioned serum, leading to reduced activation of the EGFR and its downstream signaling pathway. An antiproliferative effect appears to be derived from the increase in levels of some cytokines (i.e., TNFalpha, IL-6, OSM, and IL-8) induced by physical exercise. It is hypothesized that these molecules, in particular IL-6, released during exercise activate AMPK, which inhibits mTOR and its downstream effectors, such as p70s6k, as well as inhibits AKT and ERK1/2 phosphorylation/activation. Finally, the significant decrease in GSK3ß phosphorylation in exercise-conditioned serum highlights another possible antiproliferative effect of physical activity through the inhibition of the Wnt/ß-catenin pathway. MST1/2, Mammalian STE20-like 1/2; LATS1/2, Large Tumor Suppressor 1/2; YAP, Yes-associated protein; IGFBP1, insulin-like growth factor binding protein 1; IGF1, insulin-like growth factor; IGF-IR, IGF1 receptor; JAK, Janus tyrosine kinase; RAS; rat sarcoma virus; AKT, protein kinase B; EGF, epidermal growth factor; EGFR, epidermal growth factor receptor; TNFalpha, tumor necrosis factor alpha; IL-6, interleukine-6; OSM, oncostatin M; IL-8, interleukine-8.

**Figure 2 biology-13-00701-f002:**
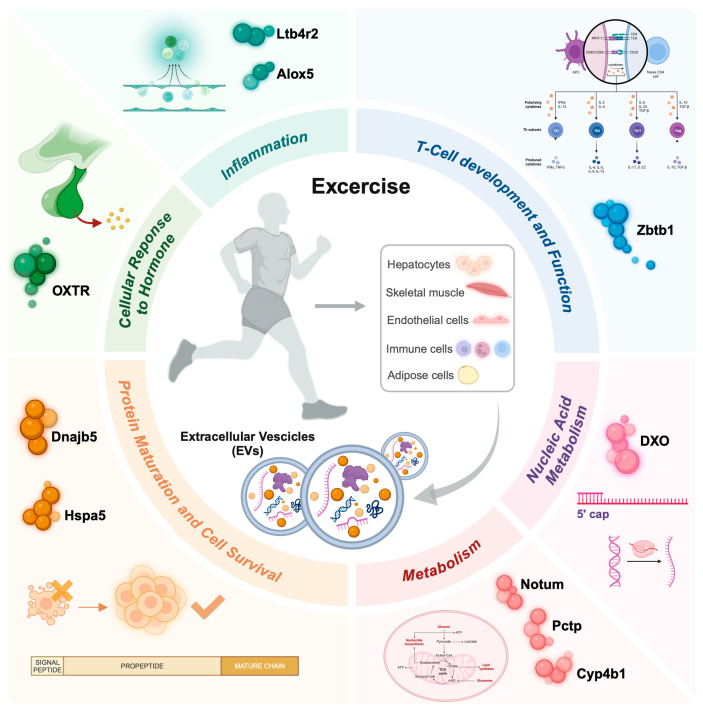
An overview depicting the impact of physical exercise on exercise-derived extracellular vesicles (EVs), emphasizing the molecular processes influenced by their cargo after exercise. EVs can be released from contracting skeletal muscles or other cell populations, particularly immune and endothelial cells, and then enter systemic circulation. Depending on their cargo (e.g., proteins, DNA, and ncRNAs), these EVs target various organs and affect several biological processes, including inflammation, immune response, cell survival, protein and nucleic acid metabolism, as well as stress and hormone responses. Notum, palmitoleoyl-protein carboxylesterase; Pctp, phosphatidylcholine transfer protein; Cyp4b1, cytochrome P450, family 4, subfamily b, polypeptide 1; Dnajb5, DnaJ Heat Shock Protein Family (Hsp40) Member B5; Hspa5, Heat Shock Protein Family A (Hsp70) Member 5; Ltb4r2, Leukotriene B4 receptor 2; Alox5, Arachidonate 5-lipoxygenase; Zbtb1, Zinc Finger And BTB Domain Containing 1; Oxtr, Oxytocin receptor; Dxo, Decapping exoribonuclease.

**Table 1 biology-13-00701-t001:** Key points related to the impact of physical activity on extracellular vesicle source, release, concentration, and size/composition.

Exercise Derived-EVs	Details
*Sources of EVs*	EVs are released by platelets, red blood cells (over 50%), and other circulating cells and tissues, including skeletal muscle (1–5%), during exercise.
*Release of EVs*	The possible contributors are as follows: - Lymphocyte mobilization. - Biomechanical forces. - Increased intracellular calcium levels. - Acidic environment. - Reactive oxygen species production.
*Concentration of EVs*	- High-intensity exercise increases circulating EVs. - Moderate-intensity exercise shows mixed results (increased, unchanged, or decreased EV levels).
*Size/composition of EVs*	- Aerobic exercise shows no change in EV size. - Resistance exercise shows mixed results (unchanged or decreased EV size).

**Table 2 biology-13-00701-t002:** Key aspects that have emerged from the treatment of cancer cells with exercise-conditioned serum.

Category	Summary
*Main Concept*	Exercise-conditioned serum inhibits crucial signaling pathways involved in cancer cell proliferation.
*Proliferation Mechanisms in Cancer*	Mutations in conserved signaling networks drive increased cancer cell proliferation.
*Impact of Exercise-Conditioned Serum*	Alters key signaling pathways, reducing cancer cell proliferation.
*Catecholamines*	Post-exercise serum is enriched in catecholamines (e.g., norepinephrine, epinephrine), which support the Hippo tumor suppressor pathway.
*Hippo Pathway Activation*	Catecholamines activate MST1/2 and MAP4K, leading to phosphorylation and activation of LATS1/2 kinases, which inhibit YAP/TAZ nuclear entry, reducing cell proliferation.
*IGFBP1*	Exercise increases IGFBP1, regulating cell proliferation and modulating IGF1 signaling pathways (JAK, RAS, AKT).
*EGF Reduction*	Lower levels of EGF reduce activation of EGFR and its downstream signaling, contributing to antiproliferative effects.
*Cytokines*	Increased levels of cytokines (TNFα, IL-6, OSM, IL-8) activate AMPK, which inhibits mTOR, AKT, and ERK1/2 pathways, reducing cell proliferation.
*GSK3β Phosphorylation*	Decreased GSK3β phosphorylation in exercise-conditioned serum inhibits the Wnt/β-catenin pathway, further contributing to antiproliferative effects.

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
