# Peer review of "Exploring the Impact of Exercise-Derived Extracellular Vesicles in Cancer Biology"

_biology, 2024, doi:10.3390/biology13090701_

Round 1

Reviewer 1 Report

Comments and Suggestions for Authors

As stated in its title “Exploring the Impact of Exercise-Derived Extracellular Vesicles in Cancer Biology”, the present review by Silvestri et al. discussed studies that focused on the biological effects of exercise-derived circulating biomolecules, with a particular focus on the role of EVs cargo, in cancer biology.

In general, the topic is interesting and of high significance. The writing was clear, engaging, and informational. The relevance of the work is well described in the introduction. The manuscript is well structured and well written.

From my point of view, the authors may add a description of the sources and the isolation methods of exercise-derived EVs in the section 2 «Extracellular vesicles and physical activity».  

Author Response

We thank the Reviewer for her/his general positive comments. We have really appreciated the suggestions from the Reviewer because he/she helped us to possibly improve the quality of our manuscript. We went through all suggestions offered by Reviewer as detailed in the point-by-point response to his/her comments.

Reviewer 1

Comments and Suggestions for Authors

As stated in its title “Exploring the Impact of Exercise-Derived Extracellular Vesicles in Cancer Biology”, the present review by Silvestri et al. discussed studies that focused on the biological effects of exercise-derived circulating biomolecules, with a particular focus on the role of EVs cargo, in cancer biology.

In general, the topic is interesting and of high significance. The writing was clear, engaging, and informational. The relevance of the work is well described in the introduction. The manuscript is well structured and well written.

Q1) From my point of view, the authors may add a description of the sources and the isolation methods of exercise-derived EVs in the section 2 «Extracellular vesicles and physical activity». 

A1) As suggest from the Reviewer, we added a description of the sources and the isolation methods of exercise-derived EVs in the section 2. We also added a Table 1, where we reported the Key points related to the impact of physical activity on extracellular vesicle source, release, concentration, and size/composition.

Reviewer 2 Report

Comments and Suggestions for Authors

I have gone through the manuscript and found this review interesting. Authors have written appropriately. Exercise derived EVs have received increased attention as they provide and opportunity to understand the mechanistic benefit of exercise in cancer patients. Emerging evidences indicates that aerobic exercise affects the circulating EVs dynamics including the size morphology and composition. Authors should also clarify the role of EVs with a focus on the dynamics of EVs in response to specific exercise modes and dosages, providing opportunities to enhance our understanding of the tailoring of exercise prescription on mediating possible cancer outcomes.

Author Response

We thank the Reviewer for her/his general positive comments. We have really appreciated the suggestions from the Reviewer because he/she helped us to possibly improve the quality of our manuscript. We went through all suggestions offered by Reviewer as detailed in the point-by-point response to his/her comments.

Reviewer 2

Comments and Suggestions for Authors

I have gone through the manuscript and found this review interesting. Authors have written appropriately. Exercise derived EVs have received increased attention as they provide and opportunity to understand the mechanistic benefit of exercise in cancer patients.

Q1) Emerging evidences indicates that aerobic exercise affects the circulating EVs dynamics including the size morphology and composition. Authors should also clarify the role of EVs with a focus on the dynamics of EVs in response to specific exercise modes and dosages, providing opportunities to enhance our understanding of the tailoring of exercise prescription on mediating possible cancer outcomes.

A1) As suggested from the Reviewer, we included in the “section 2” more information of EVs dynamics following exercise. We also added a Table 1, where we reported the Key points related to the impact of physical activity on extracellular vesicle source, release, concentration, and size/composition.

Reviewer 3 Report

Comments and Suggestions for Authors

The submitted manuscript is well written and organised and certainly worth publishing in Biology but this do not hinder its speculative spirit which is highlighted in the conclusions of the review. The reviewer would like to suggest a few points to improve.

1) in the abstract:

it is important to explain the speculative spirit of the review and to include some results on the investigation. Lines 23-27, Instead of “We will explore…” or “We will discuss…” please cite your conclusions!

2) Section 3, Biological Impact…

Give a better link between the information in the main text and the content of figure 1. The caption of figure 1 is really very informative, this is a very good point but the reader cannot easily connect the text with the figure. Perhaps a table with the main aspects and the corresponding references could help.

3) lines 171-173: the conclusion is too general, could you please develop? (Perhaps the link with the table suggested in comment 2).

4) use English for the references (the months of publication are cited in Italian).

Author Response

We thank the Reviewer for her/his general positive comments. We have really appreciated the suggestions from the Reviewer because he/she helped us to possibly improve the quality of our manuscript. We went through all suggestions offered by Reviewer as detailed in the point-by-point response to his/her comments.

Reviewer 3

Comments and Suggestions for Authors

The submitted manuscript is well written and organised and certainly worth publishing in Biology but this do not hinder its speculative spirit which is highlighted in the conclusions of the review. The reviewer would like to suggest a few points to improve.

1) in the abstract:

Q1) it is important to explain the speculative spirit of the review and to include some results on the investigation.

A1) As suggested by Reviewer, we revised the abstract.

Q2) Lines 23-27, Instead of “We will explore…” or “We will discuss…” please cite your conclusions!

A2) As suggested by Reviewer, we revised the abstract.

2) Section 3, Biological Impact…

Q3) Give a better link between the information in the main text and the content of figure 1. The caption of figure 1 is really very informative, this is a very good point but the reader cannot easily connect the text with the figure. Perhaps a table with the main aspects and the corresponding references could help.

A3) We thank the Reviewer for her/his comment. To further improve the understanding of the molecular processes induced by exercise-conditioned serum, we added a Table 2 where we reported key aspects emerged from the treatment of cancer cells with exercise-conditioned serum.

Q4) lines 171-173: the conclusion is too general, could you please develop? (Perhaps the link with the table suggested in comment 2).

A4) As suggested be Reviewer, we extended our conclusion at the end of the section 3.

Q5) use English for the references (the months of publication are cited in Italian).

A5) Sorry for the mistake, we revise accordingly.

Round 2

Reviewer 3 Report

Comments and Suggestions for Authors

The reviewer recognize the effort of authors to revise their manuscript according to all the suggestions and thank the authors for that. In particular for the production of Table 2 with the key aspects. The manuscript is now acceptable for publication in Biology.